# LINEAR CONVERGENCE FOR NATURAL POLICY GRADIENT WITH LOG-LINEAR POLICY PARAMETRIZATION

## ABSTRACT

We analyze the convergence rate of the *unregularized* natural policy gradient algorithm with log-linear policy parametrizations in infinite-horizon discounted Markov decision processes. In the deterministic case, when the Q-value is known and can be approximated by a linear combination of a known feature function up to a bias error, we show that a geometrically-increasing step size yields a linear convergence rate towards an optimal policy. We then consider the sample-based case, when the best representation of the Q-value function among linear combinations of a known feature function is known up to an estimation error. In this setting, we show that the algorithm enjoys the same linear guarantees as in the deterministic case up to an error term that depends on the estimation error, the bias error, and the condition number of the feature covariance matrix. Our results build upon the general framework of policy mirror descent and extend previous findings for the softmax tabular parametrization to the log-linear policy class.

## 1 INTRODUCTION

Sequential decision-making represents a framework of paramount importance in modern statistics and machine learning. In this framework, an agent sequentially interacts with an environment to maximize notions of reward. In these interactions, an agent observes its current state $s \in \mathcal{S}$, takes an action $a \in \mathcal{A}$ according to a policy that associates to each state a probability distribution over actions, receives a reward, and transitions to a new state. Reinforcement Learning (RL) focuses on the case where the agent does not have complete knowledge of the environment dynamics.

One of the most widely-used classes of algorithms for RL is represented by policy optimization. In policy optimization algorithms, an agent iteratively updates a policy that belongs to a given parametrized class with the aim of maximizing the expected sum of discounted rewards, where the expectation is taken over the trajectories induced by the policy. Many types of policy optimization techniques have been explored in the literature, such as policy gradient methods (Sutton et al., 1999), natural policy gradient methods (Kakade, 2002), trust region policy optimization (Schulman et al.), and proximal policy optimization (Schulman et al., 2017). Thanks to the versatility of the policy parametrization framework, in particular the possibility of incorporating flexible approximation schemes such as neural networks, these methods have been successfully applied in many settings. However, a complete theoretical justification for the success of these methods is still lacking.

The simplest and most understood setting for policy optimization is the tabular case, where both the state space $\mathcal{S}$ and the action space $\mathcal{A}$ are finite and the policy has a direct parametrization, i.e. it assigns a probability to each state-action pair. This setting has received a lot of attention in recent years and has seen several developments (Agarwal et al., 2021; Xiao, 2022). Its analysis is particularly convenient due to the decoupled nature of the parametrization, where the probability distribution over the action space that the policy assigns to each state can be updated and analyzed separately for each state. This leads to a simplified analysis, where it is often possible to drop discounted visitation distribution terms in the policy update and take advantage of the contractivity property typical of value and policy iteration methods. Recent results involve, in particular, natural policy gradient (NPG) and, more generally, policy mirror descent, showing how specific choices of learning rates yield linear convergence to the optimal policy for several formulations and variations of these algorithms (Cen et al., 2021; Zhan et al., 2021; Khodadadian et al.; Xiao, 2022; Li et al., 2022; Lan, 2022; Bhandari and Russo, 2021; Mei et al., 2020).

Two of the main shortfalls of these methods are their computational and sample complexities, which depend polynomially on the cardinality of the state and action spaces, even in the case of linear convergence. Indeed, by design, these algorithms need to update at each iteration a parameter or a probability for all state-action pairs, which has an operation cost proportional to $|\mathcal{S}||\mathcal{A}|$. Furthermore, in order to preserve linear convergence in the sample-based case, the aforementioned works assume that the worst estimate ($\ell_\infty$ norm) of $Q^\pi(s, a)$—which is the expected sum of discounted rewards starting from the state-action pair $(s, a)$ and following a policy $\pi$—is exact up to a given error threshold. Without further assumptions, meeting this threshold requires a number of samples that depends polynomially on $|\mathcal{S}||\mathcal{A}|$.

A promising approach to deal with large and high-dimensional spaces that is recently being explored is that of assuming that the environment has a low-rank structure and that, as a consequence, it can be described or approximated by a lower dimensional representation. In particular, a popular framework is that of linear function approximation, which consists in assuming that quantities of interest in the problem formulation, such as the transition probability (Linear MDPs) or the action-value function $Q^\pi$ of a policy $\pi$ can be approximated by a linear combination of a certain $d$-dimensional feature function $\phi : \mathcal{S} \times \mathcal{A} \to \mathbb{R}^d$ up to a bias error $\varepsilon_{\text{bias}}$. This linear assumption reduces the dimensionality of the problem to that of the feature function. In this setting, many researchers have proposed methods to learn the best representation $\phi$ (Agarwal et al., 2020; Modi et al., 2021; Uehara et al., 2021; Zhang et al., 2022) and to exploit it to design efficient vairations of the upper confidence bound (UCB) algorithm, for instance (Jin et al., 2020; Li et al., 2021; Wagenmaker et al., 2022).

When applying the framework of linear approximation to policy optimization, researchers typically adopt the log-linear policy class, where a policy $\pi_\theta$ parametrized by $\theta \in \mathbb{R}^d$ is defined as proportional to $\exp(\theta^\top \phi)$. For this policy class, several works have obtained improvements in terms of computational and sample complexity, as the policy update requires a number of operations that scales only with the feature dimension $d$ and the estimation assumption to retain convergence rates in the sample-based setting is weaker than the tabular counterpart. In fact, theoretical guarantees for these algorithms only assume the expectation of $Q^\pi$ over a known distribution on the state and action spaces to be exact up to a statistical error $\varepsilon_{\text{stat}}$. In the linear function approximation setting meeting this assumption typically requires a number of samples that is only a function of $d$ and it does not depend on $|\mathcal{S}|$ and $|\mathcal{A}|$ Telgarsky (2022). However, a complete understanding of the convergence rate of policy optimization methods in this setting is still missing. Recent results include *sublinear* convergence rates for unregularized NPG (Agarwal et al., 2021; Qiu et al., 2021; Zanette et al., 2021; Hu et al., 2021) and linear convergence rates for *entropy-regularized* NPG with bounded updates (Cayci et al., 2021).

Our work fills the gap between the aforementioned findings and it extends the analysis and results of the tabular setting to the linear function approximation setting. In particular, we show that, under the standard assumptions on the $(\varepsilon_{\text{stat}}, \varepsilon_{\text{bias}})$-approximation of $Q^\pi$ mentioned above, a choice of geometrically-increasing step-sizes leads to linear convergence of NPG for the log-linear policy class in both deterministic and sample-based settings. Our result directly improves upon the sublinear iteration complexity of NPG previously established for the log-linear policy class by Agarwal et al. (2021) and Hu et al. (2021) and it removes the need for entropy regularization and bounded step-sizes used by Cayci et al. (2021), under the same assumptions on the linear approximation of $Q^\pi$. Moreover, we have that the number of operations needed for the policy update and the number of samples needed to preserve the convergence rate in the sample-based setting depend on the dimension $d$ of $\phi$, as opposed to the tabular setting where the same quantities depend on $|\mathcal{S}||\mathcal{A}|$. By extending the linear convergence rate of NPG from the tabular softmax parametrization to the setting of log-linear policy parametrizations, our result directly addresses the research direction outlined in the conclusion of Xiao (2022), and it overcomes the aforementioned limitations of the tabular settings.

Our analysis is based on the equivalence of NPG and policy mirror descent with KL divergence (Raskutti and Mukherjee, 2015), which has been exploited for applying mirror-descent-type analysis to NPG by several works, such as Agarwal et al. (2021); Hu et al. (2021); Cayci et al. (2021). The advantages of this equivalence are twofold. Firstly, NPG crucially ensures a simple update rule, i.e. $\log \pi_{t+1}(a|s) = \log \pi_t(a|s) + \eta_t Q^{\pi_t}(s, a)$, which in the particular case of the log-linear policy class translates into $\theta_{t+1}^\top \phi(s, a) = \theta_t^\top \phi(s, a) + \eta_t Q^{\pi_t}(s, a)$. Secondly, the mirror descent setup is particularly useful to iteratively control the updates and the approximation errors,

e.g. through tools like the three-point descent lemma (see (3) below), and to induce telescopic sums or recursions that are often used to analyze the converge rate of the last iterate.

In our work, we show how to exploit these advantages to use the linear approximation of $Q^\pi$ in the analysis and, consequently, make weaker assumptions on the accuracy of the estimation of $Q^\pi$ w.r.t. the tabular setting. While previous results for the tabular setting (Cen et al., 2021; Zhan et al., 2021; Xiao, 2022) require an $\ell_\infty$ norm bound on the estimation error, i.e. $\|\widehat{Q}^\pi - Q^\pi\|_\infty \leq \tau$, our convergence guarantee depends on the expected error of the estimate, i.e. $\mathbb{E}(\widehat{Q}^\pi(s,a) - Q^\pi(s,a))^2 \leq \varepsilon_{\text{stat}}$, where the expectation is taken w.r.t. the discounted state visitation distribution induced by the policy $\pi$ and the uniform distribution over the action space. This allows us to employ sample-efficient policy evaluation algorithms, such as temporal difference learning (Hu et al., 2021; Telgarsky, 2022), and to remove the cardinality of the state and action spaces $|\mathcal{S}||\mathcal{A}|$ from the sample complexity of the algorithm.

The paper is organized as follows. Section 2 introduces the main setting of RL, and Section 3 introduces the algorithm framework we consider. Section 4 contains the linear approximation set-up and our main result. Section 5 presents the analysis of our main result, with the conclusions outlined in Section 6.

## 2 SETTING

Consider an agent that acts in a discounted Markov Decision Process (MDP) $\mathcal{M} = (\mathcal{S}, \mathcal{A}, P, r, \gamma, \mu)$, where: $\mathcal{S}$ is the possibly infinite state space and $\mathcal{A}$ is the finite action space; $P(s'|s,a)$ is the transition probability; $r(s,a) \in [0,1]$ is the reward function; $\gamma$ is the discount factor; and $\mu$ is the starting state distribution. A policy $\pi : \mathcal{S} \times \mathcal{A} \to \mathbb{R}$ is a probability distribution over $\mathcal{A}$ that represents the probability that an agent takes action $a$ when in state $s$. At time $t$ denote the current state and action by $s_t$ and $a_t$.

For a policy $\pi$, let $V^\pi : \mathcal{S} \to \mathbb{R}$ be the respective value function, which is defined as the expected discounted cumulative reward with starting state $s_0 = s$, namely,

$$V_s^\pi = \mathbb{E}\left[\sum_{t=0}^\infty \gamma^t r(s_t, a_t) \Big| \pi, s_0 = s\right],$$

where $a_t \sim \pi(\cdot|s_t)$ and $s_{t+1} \sim P(\cdot|s_t, a_t)$. Let $V^\pi(\mu) = \mathbb{E}_{s \sim \mu} V_s^\pi$. The agent aims to find an optimal policy $\pi^\star \in \operatorname{argmax}_\pi V^\pi(\mu)$.

For a policy $\pi$, let $Q^\pi : \mathcal{S} \times \mathcal{A} \to \mathbb{R}$ be the respective action-value function, or Q-function, which is defined as the expected discounted cumulative reward with starting state $s_0 = s$ and starting action $a_0 = a$, namely,

$$Q^\pi(s,a) = \mathbb{E}\left[\sum_{t=0}^\infty \gamma^t r(s_t, a_t) \Big| \pi, s_0 = s, a_0 = a\right],$$

where $a_t \sim \pi(\cdot|s_t)$ and $s_{t+1} \sim P(\cdot|s_t, a_t)$.

Define the discounted state visitation distribution (Sutton et al., 1999)

$$d_\mu^\pi(s) = (1-\gamma)\mathbb{E}_{s_0 \sim \mu} \sum_{t=0}^\infty \gamma^t P(s_t = s|\pi, s_0),$$

and the discounted state-action visitation distribution

$$d_\rho^\pi(s,a) = (1-\gamma)\mathbb{E}_{s_0, a_0 \sim \rho} \sum_{t=0}^\infty \gamma^t P(s_t = s|\pi, s_0, a_0),$$

where the trajectory $(s_t, a_t)_{t \geq 0}$ is generated by the MDP following policy $\pi$ and $\rho$ is a distribution over $\mathcal{S} \times \mathcal{A}$. Then we can formulate the performance difference lemma (Kakade and Langfor, 2002), a tool that will prove useful in our analysis,

$$V^\pi(\mu) - V^{\bar{\pi}}(\mu) = \frac{1}{1-\gamma}\mathbb{E}_{s \sim d_\mu^{\bar{\pi}}} \sum_{a \in \mathcal{A}} Q^\pi(s,a)(\pi(a|s) - \bar{\pi}(a|s)). \tag{1}$$

## 2.1 NOTATION

We make the following definitions for ease of exposition. As to the policy, let $\pi_s := \pi(s, \cdot)$ and $\pi^t := \pi_{\theta_t}$. For two functions $f$ and $g$, denote $(f \circ g)(x, y) = f(x)g(y)$. As to the action-value function, let $Q_s^\pi := Q^\pi(s, \cdot)$ and $Q^t(s, a) := Q^{\pi_t}(s, a)$. As to the discounted visitation distributions, let $d_\mu^t := d_\mu^{\pi^t}$, $d^t = d_\mu^t \circ \mathrm{Unif}_{\mathcal{A}}$, and $d^\star := d_\mu^\star \circ \mathrm{Unif}_{\mathcal{A}}$. Lastly, denote $\mathrm{KL}_t^\star = \mathbb{E}_{s \sim d_\mu^\star} \mathrm{KL}(\pi_s^\star, \pi_s^t)$.

## 3 NATURAL POLICY GRADIENT AND MIRROR DESCENT

*Policy class* — In this work, we consider the log-linear policy parametrization (Agarwal et al., 2021). Let $\theta \in \mathbb{R}^d$ be a parameter vector and $\phi : \mathcal{S} \times \mathcal{A} \to \mathbb{R}^d$ be a feature function. Then the policy class consists of all policies of the form:

$$\pi_\theta(a|s) = \frac{\exp(\theta^\top \phi(s, a))}{\sum_{a' \in \mathcal{A}} \exp(\theta^\top \phi(s, a'))}.$$

*Natural Policy Gradient* — We formulate NPG through mirror descent. The update at time $t + 1$ is

$$\nabla h(\pi_s^{t+1}) = \nabla h(\pi_s^t) + \eta_t Q_s^t \qquad \forall s \in \mathcal{S}, \tag{2}$$

where $h(\pi_s) = \sum_{a \in \mathcal{A}} \pi(a|s) \log \pi(a|s)$, is the entropy mirror map. This is equivalent to the update

$$\pi_{t+1}(s, a) \propto \pi_t(s, a) e^{\eta_t Q^t(s, a)} \qquad \forall s, a \in \mathcal{S}, \mathcal{A},$$

or, as in Algorithm 1, to requiring that $\theta_{t+1}$ is such that

$$\theta_{t+1}^\top \phi(s, a) = \theta_t^\top \phi(s, a) + \eta_t Q^t(s, a) \qquad \forall s, a \in \mathcal{S}, \mathcal{A}.$$

In the tabular setting, we have $d = |\mathcal{S}||\mathcal{A}|$, $\phi(s, a)$ is a vector of all zeros except a one in the position assigned to $(s, a)$, and the update is equivalent to the one analyzed by (Agarwal et al., 2021). This mirror descent setup allows us to use standard mirror descent tools in the analysis (Bubeck, 2015; Hu et al., 2021; Xiao, 2022), such as three-point descent lemma

$$D_h(\pi_s, \pi_s^t) - D_h(\pi_s, \pi_s^{t+1}) - D_h(\pi_s^{t+1}, \pi_s^t) = \langle \nabla h(\pi_s^t) - \nabla h(\pi_s^{t+1}), \pi_s^{t+1} - \pi_s \rangle \quad \forall \pi_s, \tag{3}$$

which in this setting can be expressed as

$$\mathrm{KL}(\pi_s, \pi_s^t) - \mathrm{KL}(\pi_s, \pi_s^{t+1}) - \mathrm{KL}(\pi_s^{t+1}, \pi_s^t) = -\eta_t \langle Q_s^t, \pi_s^{t+1} - \pi_s \rangle \quad \forall \pi_s. \tag{4}$$

These tools, along with the performance difference lemma (1), ensure that we can control the increase of the value function for each policy update. When we only have access to an approximation $\widetilde{Q}^\pi$ of $Q^\pi$, these tools allow controlling the error of this approximation by means of simple triangle inequality arguments, making possible the incorporation of the linear function approximation framework where $Q^\pi$ is approximated by a linear combination of the feature function $\phi$.

## 4 MAIN RESULT

In this section, we present our main result on the linear convergence of NPG. We start by introducing and discussing the assumptions and the algorithm.

### 4.1 ALGORITHM AND LINEAR FUNCTION APPROXIMATION

We make the following two assumptions on the linear approximation of $Q^\pi$, which are standard in the literature (Agarwal et al., 2021; Cayci et al., 2021).

**Assumption 4.1.** *(Bias error) Define the loss function*

$$L(w, \theta, v) := \mathbb{E}_{s,a \sim v} \left[ \left( Q^{\pi_\theta}(s,a) - w^\top \phi(s,a) \right)^2 \right]$$

*and let*

$$w_t \in \operatorname*{argmin}_w L(w, \theta_t, d_\rho^t). \tag{5}$$

*Assume that $\forall t < T$ we have*

$$L(w_t, \theta_t, d^\star) \leq \varepsilon_{bias}, \qquad L(w_t, \theta_t, d_\mu^{t+1} \circ Unif_{\mathcal{A}}) \leq \varepsilon_{bias}.$$

In order to better understand the implications of Assumption 4.1, we consider the trivial upper bound (Agarwal et al., 2021)

$$L(w_t, \theta_t, d^\star) \leq \left\| \frac{d^\star}{d_\rho^t} \right\|_\infty L(w_t, \theta_t, d_\rho^t) \leq \frac{1}{1-\gamma} \left\| \frac{d^\star}{\rho} \right\|_\infty L(w_t, \theta_t, d_\rho^t).$$

This bound allows to think of $\varepsilon_{\text{bias}}$ in Assumption 4.1 as controlling two quantities of interest. The first quantity is the loss incurred by the minimizer of $L(w, \theta_t, d_\rho^t)$, that is the best approximation $\widetilde{Q}^t = w_t^\top \phi$ of $Q^t$ with respect to the squared error averaged over the distribution $d_\rho^t$. The second quantity is the shift in distribution from $d_\rho^t$ to $d^\star$ in the loss function. A similar conclusion for Assumption 4.1 can be drawn for the the distribution $d_\mu^{t+1} \circ \text{Unif}_{\mathcal{A}}$.

**Assumption 4.2.** *(Statistical error) Assume that our estimate $\widehat{w}_t$ of $w_t$ is such that*

$$\mathbb{E}_{s,a \sim d_\rho^t} \left[ \left( \langle w_t - \widehat{w}_t, \phi(s,a) \rangle \right)^2 \right] \leq \varepsilon_{stat}.$$

Assumption 4.2 concerns the statistical error incurred when solving the minimization problem in (5) and it can be used to describe the sample complexity of the algorithm. Let $\widehat{Q}^\pi(s,a) = \widehat{w}_t^\top \phi(s,a)$ be the sample-based estimate of $\widetilde{Q}^\pi$. Then Assumption 4.2 is equivalent to assuming that

$$\mathbb{E}_{s,a \sim d_\rho^t} \left[ \left( \widehat{Q}^\pi(s,a) - \widetilde{Q}^\pi(s,a) \right)^2 \right] \leq \varepsilon_{\text{stat}}.$$

Several algorithms have been shown to satisfy Assumption 4.2 with a number of samples that depends only on the dimension $d$ of $\phi$ and not on $|\mathcal{S}|$ or $|\mathcal{A}|$, such as temporal difference learning (Telgarsky, 2022). This represent an improvement over the sample complexity of tabular algorithms, where the typical assumption $\|\widehat{Q}^\pi - Q^\pi\|_\infty \leq \varepsilon_{\text{stat}}$ (Xiao, 2022; Li et al., 2022) causes the sample complexity to depend on $|\mathcal{S}||\mathcal{A}|$.

With this set-up, we can formulate NPG with linear function approximation as in Algorithm 1. At time step $t$, let $\mathcal{D}(t, \rho)$ be an oracle such that $\widehat{w}_t = \mathcal{D}(t, \rho)$ satisfies Assumption 4.2.

---

**Algorithm 1:** NPG with linear function approximation

**Input:** Learning rate schedule $(\eta_t)_{t \geq 0}$; number of iterations $T$; initialized policy $\pi^{(0)}$; distribution $\rho$; oracle $\mathcal{D}$.
**for** $t = 0, \ldots, T-1$ **do**
    Obtain $\widehat{w}_t = \mathcal{D}(t, \rho)$.
    Update

$$\theta_{t+1} = \theta_t + \eta_t \widehat{w}_t.$$

**end for**

---

**Remark 4.3.** *(Tabular setting) It is possible to recover the tabular case by setting $d = |\mathcal{S}||\mathcal{A}|$ and $\phi(s,a)$ to be a vector of all zeros except a one in the position assigned to $(s,a)$. In this case, we recover the same update as the tabular setting and we have that $\varepsilon_{bias} = 0$, as by setting $w = Q^t$ we obtain $\mathbb{E}_{s \sim \nu, a \sim Unif_{\mathcal{A}}} \left[ \left( Q^t(s,a) - w^\top \phi(s,a) \right)^2 \right] = 0$ for any distribution $\nu$.*

**Remark 4.4.** *(Linear MDPs) Another setting for which the bias error $\varepsilon_{bias}$ is equal to $0$ is that of Linear MDPs (Jin et al., 2020), where it is assumed that the transition probability distribution and the reward function can be expressed as a linear function of the feature function $\phi$. Namely, assume there exist two feature maps $\phi : \mathcal{S} \times \mathcal{A} \to \mathbb{R}^d$ and $\mu : \mathcal{S} \to \mathbb{R}^d$ and a vector $v_r \in \mathbb{R}^d$ such that*

$$P(s'|s,a) = \langle \pi(s,a), \mu(s') \rangle, \qquad r(s,a) = \langle v_r, \phi(s,a) \rangle \qquad \forall s, s' \in \mathcal{S}, a \in \mathcal{A}.$$

*If this assumption is satisfied, then we have that $\forall s \in \mathcal{S}, a \in \mathcal{A}$*

$$Q^\pi(s,a) = r(s,a) + \gamma \int_{\mathcal{S}} V^\pi(s') P(s'|s,a) ds' = \left\langle \phi(s,a), v_r + \gamma \int_{\mathcal{S}} V^\pi(s') \mu(s') ds' \right\rangle,$$

*which means that at each time step $t$ there exists a $w_t \in \mathbb{R}^d$ such that $Q^t(s,a) = \langle w_t, \phi(s,a) \rangle$ and $L(w_t, \theta_t, d_\mu^t) = 0$.*

## 4.2 LINEAR CONVERGENCE

In order to present the main result of our work, we need two additional assumptions on the distribution mismatch coefficient and the feature covariance matrix.

**Assumption 4.5.** *Assume that the distribution mismatch coefficient*

$$\nu_\mu = \frac{1}{1-\gamma} \left\| \frac{d_\mu^\star}{\mu} \right\|_\infty$$

*is finite, i.e. $\nu_\mu < \infty$.*

Assumption 4.5 is a standard assumption in the policy optimization literature (Agarwal et al., 2021; Xiao, 2022). As we will see in Theorem 4.7, the iteration complexity of Algorithm 1 depends polynomially on this term, meaning that the convergence rate is faster when the starting state distribution $\mu$ covers the whole state space.

**Assumption 4.6.** *(Relative condition number) With respect to a distribution $v$, define*

$$\Sigma_v = \mathbb{E}_{s,a \sim v} \left[ \phi(s,a) \phi(s,a)^\top \right]$$

*and assume that there exists a $\kappa < \infty$ such that*

$$\sup_{w \in \mathbb{R}^d} \frac{w^\top \Sigma_{d^\star} w}{w^\top \Sigma_\rho w} \leq \kappa, \qquad \sup_{w \in \mathbb{R}^d} \frac{w^\top (\Sigma_{d_\mu^t \circ Unif_\mathcal{A}}) w}{w^\top \Sigma_\rho w} \leq \kappa \quad \forall t \leq T.$$

Assumption 4.6 is a standard assumption in the linear function approximation literature and highlights the importance of choosing a state-action distribution $\rho$ with good coverage over the feature space, as it can be enforced by choosing the appropriate $\rho$. In fact, if $\Phi = \{\phi(s,a)|s \in \mathcal{S}, a \in \mathcal{A}\}$ is a compact set, there always exists a state-action distribution $\rho$ such that $\kappa \leq d$ (see Lemma 23 in Agarwal et al. (2021)). In general, if $\|\phi(s,a)\|_2^2 \leq B$ for all $s \in \mathcal{S}, a \in \mathcal{A}$, we have the crude bound $\kappa \leq B/\sigma_{\min}(\Sigma_\rho)$, where $\sigma_{\min}(A)$ is the minimum eigenvalue of matrix $A$.

We are now ready to state the following theorem on the linear convergence of Algorithm 1.

**Theorem 4.7.** *(Linear convergence of NPG with log-linear parametrization) Consider NPG as in Algorithm 1 and let Assumptions 4.1, 4.2, 4.5, and 4.6 hold. If the step-size schedule satisfies*

$$\eta_{t+1} \geq \frac{\nu_\mu}{\nu_\mu - 1} \eta_t \qquad \forall t,$$

*and $\eta_0 \geq \frac{1-\gamma}{\gamma} \mathrm{KL}_t^\star$, then for every $T \geq 0$ we have*

$$V^\star(\mu) - V^T(\mu) \leq \left(1 - \frac{1}{\nu_\mu}\right)^T \frac{2}{1-\gamma} + 2\nu_\mu \sqrt{\frac{|\mathcal{A}|\kappa \varepsilon_{stat}}{(1-\gamma)^3}} + 2\nu_\mu \frac{\sqrt{|\mathcal{A}|\varepsilon_{bias}}}{1-\gamma}.$$

To the best of our knowledge, Theorem 4.7 represents the first result establishing linear convergence rates for NPG with unbounded updates and without entropy regularization for the log-linear policy

class. The convergence rate has no explicit dependence on the cardinality of the state and action spaces, with the exception of the two $|\mathcal{A}|$ terms which, as already highlighted by Agarwal et al. (2021), can be removed with a path-dependent bound. In the case where $\varepsilon_{\text{bias}} = 0$ and $\varepsilon_{\text{stat}} = 0$, the theorem recovers the same convergence rate as Theorem 10 in Xiao (2022).

To obtain the sample complexity of the algorithm, we take advantage of the theory for temporal difference learning developed by Telgarsky (2022). In particular, we have that in order to satisfy Assumption 4.2 for all $T$ iterations with high probability we need $\widetilde{O}\left(\frac{T\|w_t\|_2^2}{\varepsilon_{\text{stat}}(1-\gamma)^3}\right)$ samples. Combining this quantity with the iteration complexity from Theorem 4.7, we obtain a total sample complexity of

$$\widetilde{O}\left(\frac{\|w_t\|_2^2}{\varepsilon^2(1-\gamma)^{10}}\left\|\frac{d_\mu^\star}{\mu}\right\|_\infty\right).$$

**Remark 4.8.** *(Different policy parametrizations) While our work focuses on the log-linear policy class, it is possible to extend our framework and our analysis to general function approximation schemes. Let $f_\theta : \mathcal{S} \times \mathcal{A} \to \mathbb{R}$ be a parameterized function and define the policy class $\{\pi_\theta | \theta \in \Theta\}$ as*

$$\pi_\theta(a|s) = \frac{\exp(f_\theta(s,a))}{\sum_{a'\in\mathcal{A}}\exp(f_\theta(s,a'))}.$$

*Let $g_\omega$ be a parametrized operator of $f_\theta$, define the loss function*

$$L(\omega,\theta,\nu) := \mathbb{E}_{s\sim\nu,a\sim\text{Unif}_\mathcal{A}}\left[(Q^{\pi_\theta}(s,a) - g_\omega(f_{\theta_t}(s,a)))^2\right]$$

*and let*

$$\omega_t \in \underset{\omega}{\arg\min}\, L(\omega,\theta_t,d_\mu^t).$$

*Then, Assumption 4.1 becomes*

$$L(\omega_t,\theta_t,d_\mu^\star) \leq \varepsilon_{bias} \qquad \forall t < T.$$

*The NPG update* (2) *can then be formulated as requiring $\theta_{t+1}$ to be such that*

$$f_{\theta_{t+1}}(s,a) = f_{\theta_t}(s,a) + \eta_t g_{\omega_t}(f_{\theta_t}(s,a)) \qquad \forall s \in \mathcal{S}, a \in \mathcal{A}.$$

*Finding methods to solve this system of equations is beyond the scope of this work.*

## 5 ANALYSIS

In order to prove Theorem 4.7, we need some intermediate results. The first one regards the decomposition of the statistical and the bias errors.

**Lemma 5.1.** *The expected error of the estimate $\widehat{Q}_s^t$ of $Q_s^t$ can be bounded as follows*

$$\left|E_{s\sim v}\langle Q_s^t - \widehat{Q}_s^t, \pi_s^t - \pi_s\rangle\right| \leq 2\sqrt{\frac{|\mathcal{A}|\kappa\varepsilon_{stat}}{1-\gamma}} + 2\sqrt{|\mathcal{A}|\varepsilon_{bias}} \qquad \forall t < T,$$

*for both $v = d_\mu^{t+1}, \pi_s = \pi_s^{t+1}$ and $v = d_\mu^\star, \pi_s = \pi_s^\star$.*

*Proof of Lemma 5.1.* We start by adding and subtracting $\widehat{Q}_s^t$

$$\left|E_{s\sim v}\langle Q_s^t - \widehat{Q}_s^t, \pi_s^t - \pi_s^{t+1}\rangle\right| \leq \left|\mathbb{E}_{s\sim v}\langle\widetilde{Q}_s^t - \widehat{Q}_s^t, \pi_s^t - \pi_s\rangle\right| + \left|\mathbb{E}_{s\sim v}\langle Q_s^t - \widetilde{Q}_s^t, \pi_s^t - \pi_s\rangle\right|.$$

We then bound the two terms on the right-hand side separately. For the first term, we have that

$$\left|\mathbb{E}_{s\sim v}\langle\widetilde{Q}_s^t - \widehat{Q}_s^t, \pi_s^t - \pi_s\rangle\right|$$

$$\leq \left|\mathbb{E}_{s\sim v,a\sim\pi_s^t}\left[(w_t - \widehat{w}_t)\phi(s,a)\right]\right| + \left|\mathbb{E}_{s\sim v,a\sim\pi_s}\left[(w_t - \widehat{w}_t)\phi(s,a)\right]\right|$$

$$\leq \sqrt{\mathbb{E}_{s\sim v,a\sim\pi_s^t}\left[((w_t - \widehat{w}_t)\phi(s,a))^2\right]} + \sqrt{\mathbb{E}_{s\sim v,a\sim\pi_s}\left[((w_t - \widehat{w}_t)\phi(s,a))^2\right]}$$

$$\leq 2\sqrt{|\mathcal{A}|\mathbb{E}_{s\sim v,a\sim\text{Unif}_\mathcal{A}}\left[((w_t - \widehat{w}_t)\phi(s,a))^2\right]} = 2\sqrt{|\mathcal{A}|\,\|w_t - \widehat{w}_t\|_{\Sigma_{v\circ\text{Unif}_\mathcal{A}}}^2},$$

where $\|w\|_\Sigma^2 := w^\top \Sigma w$. Using Assumption 4.6 and the fact that $(1 - \gamma)\rho \le d_\rho^t$ we have

$$\|w_t - \widehat{w}_t\|_{\Sigma_{v \circ \mathrm{Unif}_\mathcal{A}}}^2 \le \kappa \|w_t - \widehat{w}_t\|_{\Sigma_\rho}^2 \le \frac{\kappa}{1 - \gamma} \|w_t - \widehat{w}_t\|_{\Sigma_{d_\rho^t}}^2 \le \frac{\kappa \varepsilon_{\mathrm{stat}}}{1 - \gamma}.$$

Similarly, for the second term we have

$$\left| \mathbb{E}_{s \sim v} \langle Q_s^t - \widetilde{Q}_s^t, \pi_s^t - \pi_s \rangle \right| \le 2\sqrt{|\mathcal{A}| \mathbb{E}_{s \sim v, a \sim \mathrm{Unif}_\mathcal{A}} \left[ \left( Q^t(s, a) - \widetilde{Q}^t(s, a) \right)^2 \right]} \le 2\sqrt{|\mathcal{A}| \varepsilon_{\mathrm{bias}}}.$$

$\square$

For ease of exposition, in the rest of this section denote

$$\tau := 2\sqrt{\frac{|\mathcal{A}| \kappa \varepsilon_{\mathrm{stat}}}{1 - \gamma}} + 2\sqrt{|\mathcal{A}| \varepsilon_{\mathrm{bias}}}.$$

The next lemma regards the quasi-monotonic improvements of Algorithm 1. Let $\widehat{Q}_s^t = \widehat{w}_t^\top \phi(s, \cdot)$.

**Lemma 5.2.** *The updates of Algorithm 1 satisfy, for all $s \in \mathcal{S}$,*

$$\langle \widehat{Q}_s^t, \pi_s^{t+1} - \pi_s^t \rangle \ge 0$$

*and*

$$V^{t+1}(\mu) - V^t(\mu) \ge -\frac{\tau}{1 - \gamma}.$$

*Proof of Lemma 5.2.* By 1-strong convexity of $h$ on $(0, 1)$, we have $\forall s \in \mathcal{S}$

$$0 \le \left\| \pi_s^{t+1} - \pi_s^t \right\|_2^2 \le \langle \nabla h(\pi_s^{t+1}) - \nabla h(\pi_s^t), \pi_s^{t+1} - \pi_s^t \rangle = \langle \widehat{Q}_s^t, \pi_s^{t+1} - \pi_s^t \rangle.$$

As to the second inequality, we use the performance difference lemma (1) and Lemma 5.1 to obtain

$$\begin{aligned} (1 - \gamma)(V^{t+1}(\mu) - V^t(\mu)) &= \mathbb{E}_{s \sim d_\mu^{t+1}} \langle Q_s^t, \pi_s^{t+1} - \pi_s^t \rangle \\ &= \mathbb{E}_{s \sim d_\mu^{t+1}} \langle \widehat{Q}_s^t, \pi_s^{t+1} - \pi_s^t \rangle + \mathbb{E}_{s \sim d_\mu^{t+1}} \langle Q_s^t - \widehat{Q}_s^t, \pi_s^{t+1} - \pi_s^t \rangle \\ &\ge -\frac{\tau}{1 - \gamma}. \end{aligned}$$

$\square$

The last result we need is the following lemma, which can be straightforwardly proven by induction.

**Lemma 5.3.** *Suppose $0 < \alpha < 1, b > 0$ and a nonnegative sequence $\{a_k\}$ satisfies*

$$a_{k+1} \le \alpha a_k + b \qquad \forall k \ge 0.$$

*Then for all $k \ge 0$,*

$$a_k \le \alpha^k a_0 + \frac{b}{1 - \alpha}.$$

With these results in place, we are ready to prove Theorem 4.7.

*Proof of Theorem 4.7.* Let

$$\nu_k = \left\| \frac{d^\star}{d_\mu^{t+1}} \right\|_\infty$$

and consider the equality in (4)

$$\mathrm{KL}(\pi_s, \pi_s^t) - \mathrm{KL}(\pi_s, \pi_s^{t+1}) - \mathrm{KL}(\pi_s^{t+1}, \pi_s^t) = -\eta_t \langle Q_s^t, \pi_s^{t+1} - \pi_s \rangle \quad \forall \pi_s.$$

Then, for $\pi_s = \pi_s^\star$ we have that

$$\mathbb{E}_{s \sim d_\mu^\star} \langle \widehat{Q}_s^t, \pi_s^t - \pi_s^{t+1} \rangle + \mathbb{E}_{s \sim d_\mu^\star} \langle \widehat{Q}_s^t, \pi_s^\star - \pi_s^t \rangle \le \mathrm{KL}_t^\star - \mathrm{KL}_{t+1}^\star. \tag{6}$$

We bound the two terms on the left-hand side separately. For the first one, we have that

$$
\mathbb{E}_{s \sim d_\mu^\star} \langle \widehat{Q}_s^t, \pi_s^t - \pi_s^{t+1} \rangle
$$
$$
\geq \left\| \frac{d^\star}{d_\mu^{t+1}} \right\|_\infty \mathbb{E}_{s \sim d_\mu^{t+1}} \langle \widehat{Q}_s^t, \pi_s^t - \pi_s^{t+1} \rangle
$$
$$
= \nu_{k+1}(1-\gamma) \left( V^t(\mu) - V^{t+1}(\mu) \right) + \nu_{k+1} \mathbb{E}_{s \sim d_\mu^{t+1}} \langle \widehat{Q}_s^t - Q_s^t, \pi_s^t - \pi_s^{t+1} \rangle
$$
$$
\geq \nu_{k+1}(1-\gamma) \left( V^t(\mu) - V^{t+1}(\mu) \right) - \nu_{k+1}\tau,
$$

where the first inequality is due to Lemma 5.2, the equality is due to the performance difference lemma (1) and the second inequality is due to Lemma 5.1. We use Lemma 5.1 again to bound the second term in the left-hand side of (6)

$$
\mathbb{E}_{s \sim d_\mu^\star} \langle \widehat{Q}_s^t, \pi_s^\star - \pi_s^t \rangle = \mathbb{E}_{s \sim d_\mu^\star} \langle Q_s^t, \pi_s^\star - \pi_s^t \rangle + \mathbb{E}_{s \sim d_\mu^\star} \langle \widehat{Q}_s^t - Q_s^t, \pi_s^\star - \pi_s^t \rangle
$$
$$
\geq (1-\gamma) \left( V^\star(\mu) - V^t(\mu) \right) - \tau.
$$

Plugging the two bounds in (6) we obtain

$$
\nu_{k+1} \left( \Delta_{t+1} - \Delta_t - \frac{\tau}{1-\gamma} \right) + \Delta_t \leq \frac{\mathrm{KL}_t^\star}{(1-\gamma)\eta_t} - \frac{\mathrm{KL}_{t+1}^\star}{(1-\gamma)\eta_t} + \frac{\tau}{1-\gamma},
$$

where $\Delta_t = V^\star(\mu) - V^t(\mu)$. From Lemma 5.2 we have that $\Delta_{t+1} - \Delta_t - \frac{\tau}{1-\gamma} \leq 0$, so, since $\nu_{t+1} \leq \nu_\mu$, we can replace $\nu_{t+1}$ with $\nu_\mu$ and write

$$
\nu_\mu \left( \Delta_{t+1} - \Delta_t \right) + \Delta_t \leq \frac{\mathrm{KL}_t^\star}{(1-\gamma)\eta_t} - \frac{\mathrm{KL}_{t+1}^\star}{(1-\gamma)\eta_t} + \frac{(1+\nu_\mu)\tau}{1-\gamma}.
$$

Rearranging and dividing by $\nu_\mu$ we obtain

$$
\Delta_{t+1} + \frac{\mathrm{KL}_{t+1}^\star}{(1-\gamma)\nu_\mu \eta_t} \leq \left( 1 - \frac{1}{\nu_\mu} \right) \left( \Delta_t + \frac{\mathrm{KL}_t^\star}{(1-\gamma)\eta_t(\nu_\mu - 1)} \right) + \left( 1 + \frac{1}{\nu_\mu} \right) \frac{\tau}{1-\gamma}.
$$

If the step sizes satisfy $\eta_{t+1}(\nu_\mu - 1) \geq \eta_t \nu_\mu$, then

$$
\Delta_{t+1} + \frac{\mathrm{KL}_{t+1}^\star}{(1-\gamma)\eta_{t+1}(\nu_\mu - 1)} \leq \left( 1 - \frac{1}{\nu_\mu} \right) \left( \Delta_t + \frac{\mathrm{KL}_t^\star}{(1-\gamma)\eta_t(\nu_\mu - 1)} \right) + \frac{2\tau}{1-\gamma}
$$

where we used that $\nu_\mu \geq 1$. The proof of the theorem follows by applying Lemma 5.3. $\qquad\square$

## 6 CONCLUSION

We show how unregularized NPG can be tuned to achieve linear convergence for the log-linear policy class up to an error floor that depends on the statistical error of our estimates of $Q^t$ and the bias error of the best linear approximation of $Q^t$. Our results fill the gap between the findings in the tabular setting and the log-linear policy setting, taking advantage of a mirror-descent type analysis, and address research directions outlined in previous works (Xiao, 2022). The main future direction is that of extending our framework and results to general policy parametrizations, as we suggest in Remark 4.8.

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
