# OpenReview forum: "Linear convergence for natural policy gradient with log-linear policy parametrization"
_ICLR.cc/2023/Conference — Submitted to ICLR 2023_

### Official Review · Reviewer_vAhQ · 2022-10-23

**Confidence:** 4
**Correctness:** 4
**Technical Novelty And Significance:** 2
**Empirical Novelty And Significance:** 2
**Recommendation:** 6

**Clarity, Quality, Novelty And Reproducibility:**

The author may need to discuss in what scenario their established result really outperforms the pervious SOTA results of NPG. After doing that, readers are able to justify the real contribution of this paper.

**Strength And Weaknesses:**

Strength:
(1) This paper is well-written and easy to follow
(2) The technique proof is solid and clear
(3) The result established in this paper is different from all previous works in the same setting, which is interesting.

Weakness:
(1) The major technique contribution of this paper is in exploring how the incremental increasing stepsize with the special RL optimization nature can provide a linear convergence rate, which is interesting but not strong enough.
(2) The linear convergence rate in this paper may not be very useful in most of the case. The major reason is that the contraction factor $(1-1/\nu_\mu)$ is significantly larger than \gamma (or very close to 1 as $\nu_\mu$ is usually very large. With a given sample complexity $\epsilon$, if $\nu_\mu$ scale as $\mathcal{O}(\exp(1/\epsilon))$ then the sample complexity in this paper may actually looser than standard NPG complexity results established in many previous works. Thus, the linear convergence rate established in this paper only improves the real sample complexity of NPG within a very limited scenario.

**Summary Of The Paper:**

This paper focuses on the NPG in softmax policy with linear function approximation. By proposing a new updating strategy, in which the stepsize increases at a designed rate, this paper is able to show that the NPG enjoys a per-iteration geometry convergence rate w.r.t a factor $1-1/\nu_\mu$. Since previous works of NPG in the similar setting mostly focus on establishing sublinear convergence rate, the result established in this paper is interesting.

**Summary Of The Review:**

Overall this paper provide a very interesting result of NPG. However, the result is more like provide a different formulation of the sample complexity result of NPG instead of establishing a more tighter bound (for detail see my comment in weakness in previous section).


============ post rebuttal ============


After reviewing author's response, I will raise my score to 6. The contribution in this paper is not strong enough given the state-of-the-art. The linear convergence rate of PG without entropy regularization has been explored in previous work. For example, in [1], when adopting l2 regularization in their mirror descent update, the result in [1] naturally implies a linear convergence rate for PG algorithm. However, since this paper explore a new technique for analyzing the PG type algorithm, it might be interesting to the community.

[1] Lan, G. (2022). Policy mirror descent for reinforcement learning: Linear convergence, new sampling complexity, and generalized problem classes. Mathematical programming, 1-48.

---

> ### Author Response · Authors · 2022-11-18
> **Answer to Reviewer vAhQ**
>
> We thank the reviewer for their valuable comments. Please refer to the general comment and see below for an answer to the issues you have mentioned.
>
> **[Convergence rate]** Please refer to the **[Distribution mismatch coefficient]** paragraph in the general comment for an answer to the reviewer's concerns on the convergence rate.
>
> **[Contribution]** The main focus of this work is the finite time convergence analysis for natural policy gradient with log-linear classes. This type of results are typical of the literature on policy optimization, e.g. [Mei et al., 2020; Cayci et al., 2021; Cen et al., 2021; Zhan et al., 2021; Bhandari and Russo, 2021; Xiao, 2022], and are of interest as they help to understand the path of the algorithm.

---

### Official Review · Reviewer_VEtG · 2022-10-24

**Confidence:** 4
**Correctness:** 3
**Technical Novelty And Significance:** 2
**Empirical Novelty And Significance:** Not applicable
**Recommendation:** 5

**Clarity, Quality, Novelty And Reproducibility:**

In general the paper is well written and has good clarity. However, the technical novelty seems a bit limited. It is a theory paper so there is no reproducibility issue.

**Strength And Weaknesses:**

Strength: The paper developed new results on natural PG method with log-linear parameterization, and is well-written generally (up to some small errors). The motivation is clear, and the manuscript is concise and easy to follow.

Weakness: As a pure theory paper, I am a bit uncertain about the technical novelty and solidness of the results. It seems that it strongly relies on the analysis in previous works, e.g., Agarwal et al, 2021, Xiao 2022, and Telgarsky et al., 2022. The idea of increasing the stepsize also follows from Xiao 2022. I skimmed through the proof, which reinforces my feeling as well.

Other Comments:
1. As it is a theory paper with new technical results, it would be helpful to summarize the novelty of the "Techniques" in the front, e.g., in introduction, so that the readers can better understand it and compare it with the literature.
2. It would be better if some simple experiments can be provided to corroborate the theoretical findings: how does the stepsize affect the convergence rates, how does it compare with the existing PG algorithms, and any ablation study about the algorithm parameters (stpesize, feature vector, etc.).
3. It was claimed in the abstract and intro multiple times that the results do not depend on |S|, |A| but only on d. However, in the main results in Theorem 4.7, there is still dependence on |A|. It was argued that "can be removed with a path-dependent bound". It was not very convincing to me to claim this without a valid statement. It is better to either give a formal statement here, or modify the claims in the beginning.
4. Errors and typos:
1) Definition of $d_\rho^\pi(s,a)$ on page 3 is not correct.
2) Some notation abbreviation in sec. 2.1 is not consistent with later sections, e.g., $\pi^t$ v.s. $\pi_t$. Where is $d_\mu^*$ defined? Since it is a theory paper, being rigorous about notation would be important and affect the credential of the paper.
3) There is an additional "the" in the sentence right before Assumption 4.2.
4) Definition of linear MDP in Remark 4.4 is not correct: P and r should be sharing the same feature $\phi$.


**Post-rebuttal:**

Thank the authors for the rebuttal. I have read it, and also read other reviewers' comments. Now I get that the mirror descent reformulation of the NPG update in the linear function approximation case is new. But I am still a bit unsure if the overall novelty clears the bar of ICLR. I recommend the authors to maybe add the results regarding neural network and even general function parameterization more formally, to make the results more complete and stronger. I will hence stick to my previous evaluation. Thank you.

**Summary Of The Paper:**

This paper analyzes the convergence rate of the natural policy gradient (PG) algorithm with log-linear policy parametrizations in infinite-horizon discounted MDPs. The paper shows that under standard function approximation assumption about the Q-value functions, with an increasing stepsize, a linear convergence rate of the natural PG method can be obtained for log-linear policy class in both deterministic and sample-based settings. The results strictly improve the existing results along this direction, which can only have either sublinear convergence rate, or require regularization and bounded iterate assumptions. In the sample-based setting, the results do not depend on the cardinality of the spaces, but on the dimension of the features, which is favorable for large-scale cases. The paper is clearly written in general, and the results are of interest to the community.

**Summary Of The Review:**

In sum, the paper presented some interesting results about natural policy gradient for log-linear policy classes. The techniques and idea about the algorithm (increasing the stepsize specifically), are based on several most related literature, making the novelty and orginality of the work a bit limited to make the cut.

---

> ### Author Response · Authors · 2022-11-18
> **Answer to Reviewer VEtG**
>
> We thank the reviewer for their valuable comments. Please refer to the general comment and see below for an answer to the issues you have mentioned, in particular **[Main contribution]**, **[Better sample complexity]** and **[General policy parametrizations]**.
>
> **[Technique]** The main technique introduced by this paper is the mirror descent formulation of Natural Policy Gradient for non-tabular settings. This is substantially different from the analysis in [Agarwal et al., 2021], where the analysis for the function approximation setting relies heavily on assumptions on smoothness and bounded updates. Using a mirror descent setup, we are able to control the updates through the three point descent lemma. As we highlight in Remark 4.8, this interpretation of NPG holds for any kind of parametrization and represents a theoretically sound alternative to SOTA policy gradient methods for policies parametrized by neural networks. We will add these remarks to the discussion at the end of page 2 and beginning of page 3.
>
> **[Experiments]** The scope of this work is the theoretical analysis of natural policy gradient through mirror descent techniques. As highlited in the general comment, we believe that our work paves the way for new experimental results that are, however, beyond the scope of this paper.
>
> **[Dependence on $|\mathcal{A}|$]** We agree with the reviewer. We will revise the paper to either include a formal statement or revise the claim.
>
> **[Errors and typos]** We thank the reviewer for spotting these, we will correct them.

---

### Official Review · Reviewer_5t57 · 2022-10-25

**Confidence:** 4
**Correctness:** 4
**Technical Novelty And Significance:** 2
**Empirical Novelty And Significance:** Not applicable
**Recommendation:** 5

**Clarity, Quality, Novelty And Reproducibility:**

**Quality**: Technical difficulty is limited since this work is using one existing technique into another one to obtain a not surprising result.

**Clarity**: The paper is well written and the results are clearly presented.

**Originality**: Originality is limited as mentioned above.

**Strength And Weaknesses:**

**Strength**

The linear convergence rate was previously shown in softmax tabular parametrization, and this work shows that the same rate can be achieved with log-linear parametrization, which was not known before.

**Weaknesses**

1. The above mentioned results are new, but the novelty and technical contributions of this work are very limited to my knowledge for the following reasons.
    - The main analysis is based on decomposing $V^*(\mu) - V^T(\mu)$ into two parts of optimization error and biases, which has already been shown by Agarwal et al., 2021. This work adopted the same assumptions in Agarwal et al., 2021, i.e., Assumptions 4.1, 4.2, 4.5, and 4.6.
    - Using the decomposition of $V^*(\mu) - V^T(\mu)$, stepsize in Xiao, 2022 is then used.
    - The analyses also look very similar to those two works to me.

**Summary Of The Paper:**

This work proposes to use increasing stepsize in Xiao, 2022, as shown in Theorem 4.7, in natural policy gradient with log-linear policy parametrization, and the main result is that the convergence rate is linear plus approximation errors, improving the $O(1/\sqrt{T})$ rate plus approximation error terms in Agarwal et al, 2021.


**Summary Of The Review:**

Overall, this work combines two existing techniques of Agarwal et al., 2021 and Xiao, 2022, and obtains a new linear convergence rate for natural policy gradient with log-linear parametrization. The results are correct and new. However, to my understanding, the technical difficulty and novelty of this work are limited.


====Update====

Thank you for the comments. However, after reading the authors' feedback and other reviewers' comments, I would keep my score.

As mentioned by the authors in the feedback, the main introduced technique is "the mirror descent formulation of Natural Policy Gradient for non-tabular settings", which "is substantially different from the analysis in [Agarwal et al., 2021]".

The "mirror descent formulation of Natural Policy Gradient" is actually not new. For example, the Q-NPG method in Agarwal2021 is similar to Eq. (2) and Algorithm 1 in this paper.

My understanding of the authors' claim is that the authors used the analysis in Xiao2022 to the settings in Agarwal2021. The analysis of this paper (performance difference lemma and the three point descent lemma) contains also widely used techniques in mirror decent proofs. The authors' feedback still confirms my impression of incremental technical contribution.

If the authors wanted to emphasize more the technical innovation, I suggest comparing and illustrating the difference and difficulty more directly, such that the audience can better understand the contributions of this work.

---

> ### Author Response · Authors · 2022-11-18
> **Answer to Reviewer 5t57**
>
> We thank the reviewer for their consideration. Please refer to the general comment for an answer to the issues you have mentioned, in particular **[Main contribution]**, **[Better sample complexity]** and **[General policy parametrizations]**.

---

### Official Review · Reviewer_SUKm · 2022-10-25

**Confidence:** 4
**Clarity, Quality, Novelty And Reproducibility:** See "Strength and Weakness".
**Correctness:** 3
**Technical Novelty And Significance:** 3
**Empirical Novelty And Significance:** Not applicable
**Recommendation:** 5

**Strength And Weaknesses:**

Strength:

- This work investigates the problem of policy optimization with log-linear policy parameterization, which is a fundamental problem of the policy optimization theory.
- This work extends the insight of [Xiao, 2022] from the tabular setting to the regime of linear function approximation and manages to achieve a linear converging error bound (up to some statistical and approximation errors) without introducing entropy regularization [Cayci et al., 2021].

Weaknesses:

- Assumption 4.6 imposes additional bound on the relative condition number for all iterations (compared to Assumption 6.2 in [Agarwal et al., 2021]), which is somewhat undesirable. It would be great if the authors can make this clearer when making a comparison and provide a justification.
- The claim "convergence towards an optimal policy" appears to lack some support. In particular, it remains unclear if the bound established in Theorem 4.7 is tight when compared with an optimal policy. In addition, the bound introduce a factor $\nu_\mu$ in the dependency on $\epsilon_{stat}$ and $\epsilon_{bias}$ (which is at least $1/(1-\gamma)$) compared with Theorem 6.1 in [Agarwal et al., 2021]. This makes the linear converging bound less appealing.

**Summary Of The Paper:**

This work studies policy optimization in single-agent RL with log-linear policy parameterization by focusing on natural policy gradient (NPG) method. In particular, this work demonstrates a linear converging error bound of NPG with increasing step sizes to the optimal policy, up to some statistical and approximation errors.

**Summary Of The Review:**

This work makes interesting contribution towards policy optimization with log-linear policy optimization, by extending the insight of [Xiao, 2022] which focused on the tabular setting. This work establishes a linear converging upper bound up to an error floor determined by $\epsilon_{stat}$ and $\epsilon_{bias}$, by deploying NPG updates with increasing step sizes. However, the magnitude of the error floor is worse than that of Theorem 6.1 in [Agarwal et al., 2021], despite introducing additional assumptions. Therefore, the reviewer vote for "5: marginally below the acceptance threshold".

---

> ### Author Response · Authors · 2022-11-18
> **Answer to Reviewer SUKm**
>
> We thank the reviewer for their valuable comments. Please refer to the general comment and see below for an answer to the issues you have mentioned.
>
> **[Relative condition number]** There are two justifications for Assumption 4.6. The first and more intuitive one is that we need a starting state distribution that sufficiently covers the feature space with respect to both the optimal policy and the current policy, since we need to evaluate the current policy in order to perform the update. The second and more rigorous justification is that if $\left\lVert \phi(s,a) \right\rVert_2^2\leq B$ for all $s\in\mathcal{S},a\in\mathcal{A}$, which is a standard requirement, we can guarantee $\kappa\leq dB$.
>
> Proof: Let $\Phi = \{\phi(s,a)|s\in\mathcal{S},a\in\mathcal{A}\}$, then we can apply Theorem 3 in [Bubeck et al., 2012] (arXiv:1202.3079) and obtain that there exists a distribution $\rho$ supported on $M\leq d(d + 1)/2 + 1$ points $\phi_1,\dots,\phi_M$ in $\Phi$ such that $w^\top = d\sum_{i = 1}^M\rho_i w^\top\phi_i\phi_i^\top = d w^\top \Sigma_\rho$. Then, for any distribution $\nu$, we have that $$\sup_{w\in\mathcal{R}^d}\frac{w^\top\Sigma_{\nu}w}{w^\top\Sigma_{\rho}w} = d\sup_{w\in\mathcal{R}^d}\frac{w^\top\Sigma_{\nu}w}{w^\top w}\leq dB.$$
>
> **[Optimal policy]** $\pi^\star$ is, as the reviewer suggests, a generic comparator policy, that needs not be necessarily the optimal policy. We thank the reviewer for pointing this out, we will amend this in the paper.

---

### Official Review · Reviewer_XCzT · 2022-10-27

**Confidence:** 5
**Correctness:** 4
**Technical Novelty And Significance:** 2
**Empirical Novelty And Significance:** Not applicable
**Recommendation:** 3

**Clarity, Quality, Novelty And Reproducibility:**

- In Remark 4.4 I believe by \pi you mean \phi.
- The sample complexity stated right before Remark 4.8 is not clear. What is this sample complexity representing? What is w_t here? This is sample complexity of reaching epsilon neighborhood of what value?

**Strength And Weaknesses:**

Strength:
The paper extends the linear convergence rate shown for tabular setting in Xiao (2022) to a more general function approximation setting.
Weakness:
- In the bound, v_\mu can be in general state and action dependent. Also it depends on 1/(1-\gamma). The authors do not specify the dependency of this parameter on state, action and 1/(1-\gamma). Since this parameter appears in an exponential manner in the upper bound, this can result in a very bad dependency on the upper bound.
- The contribution of the paper is limited compared to the prior work, specially Xiao (2022).
- The error terms appearing on the upper bound of Theorem 4.7 can be very loose. In particular, we have a very bad dependency in terms of 1/(1-\gamma).

**Summary Of The Paper:**

This paper studies the geometric convergence of NPG parametrized by log linear function. The authors assume that approximation of Q-functions by linear function, and approximation of policy with log-linear function has some error bound, and they show that employing NPG with an increasing step size result in exponential convergence to a ball around the global optimum. This ball is proportional to these function approximation errors.

**Summary Of The Review:**

I believe the contribution of the paper is limited, and is a straightforward extension of the prior work.
Also the paper is not well written. Some parts are not clear. There is a sample complexity which is not clear corresponds to what.

---------------------------------------------------------------------------------

Update: After looking at the response by the authors, I would like to still keep my score. I believe the paper is a simple extension of the Xiao (2022) to the function approximation setting.

---

> ### Author Response · Authors · 2022-11-18
> **Answer to Reviewer XCzT**
>
> We thank the reviewer for their valuable comments. Please refer to the general comment and see below for an answer to the issues you have mentioned.
>
> **[Sample complexity]** The sample complexity before remark 4.8 refers to the number of samples needed for the algorithm to converge to $V^T(\mu)$ such that $V^\star(\mu)-V^T(\mu)\leq \varepsilon + 2\nu_\mu\frac{\sqrt{|\mathcal{A}|\varepsilon_\text{bias}}}{1-\gamma}$. We thank the reviewer for pointing out that this was unclear, we will revise it.

---

### Author Response · Authors · 2022-11-18
**Gereral comment**

We thank the reviewers for their consideration and helpful remarks. We address several common concerns below.

**[Main contribution]** The main contribution of this work is the extention of previous results on convergence rates for the tabular setting to different types of parametrization. This contribution, which was notably stated as a future direction of research by [Mei et al., 2020; Cen et al., 2021; Zhan et al., 2021; Bhandari and Russo, 2021; Xiao, 2022], is the next natural stepping stone for understanding the experimental results of policy optimization methods. The proof of Theorem 4.7 requires a careful analysis, due to the fact that the results on policy mirror descent from [Xiao, 2022] and the results on compatible function approximation from [Agarwal et al., 2021] can not be combined without a well-chosen mirror descent formulation of natural policy gradient for general policy classes (see equation (2)).


**[Better sample complexity]** In addition to the extention of the convergence result, we make weaker assumptions on the accuracy of the estimate $\widehat{Q}^\pi(s,a)$ than [Xiao, 2022]: while he requires $|\widehat{Q}^\pi-\widetilde{Q}^\pi|_\infty\leq\varepsilon_\text{stat}$, we only require $\mathbb{E}[(\widehat{Q}^\pi-\widetilde{Q}^\pi)^2]\leq\varepsilon_\text{stat}$. Therefore, the number of samples needed at each update to satisfy the assumptions in [Xiao, 2022] depend on $|\mathcal{S}||\mathcal{A}|$, whereas the number of sample needed to satisfy our assumptions depends only on the feature vector $\phi$.


**[General policy parametrizations]** While our paper focuses on the setting of linear parametrization, the same analysis holds for general parametrizations, such as neural networks (NNs), as we specify in Remark 4.8.
More precisely, let $f_\theta:\mathcal{S}\times\mathcal{A}\rightarrow\mathcal{R}$ and $g_\omega:\mathcal{S}\times\mathcal{A}\rightarrow\mathcal{R}$ be parameterized functions, e.g.\ NNs, define the policy class $\\{\pi_\theta|\theta\in\Theta\\}$ as
$$\pi_\theta(a|s) = \frac{\exp(f_\theta(s,a))}{\sum_{a'\in\mathcal{A}}\exp(f_\theta(s,a'))}$$
and define the loss function
$$L(\omega,\theta,\nu) := \mathbb{E}\_{s\sim \nu, a\sim \text{Unif}\_\mathcal{A}}\left[\left(Q^{\pi_\theta} (s,a)- g_\omega(s,a)\right)^2\right].$$
Then it is possible to design an actor-critic algorithm where the critic approximates the Q-function, that is finds
$$\omega_t\in\mathop{\mathrm{argmin}}\_\omega L(\omega,\theta_t,d^t_\rho),$$
and where the actor updates the policy solving the minimization problem $$\theta_{t+1}\in\mathop{\mathrm{argmin}}\_\theta
\left\lVert f_\theta - f_{\theta_t} - \eta_t g_{\omega_t} \right\rVert_2.$$
The third contribution is then providing an insight on policy gradient methods used in practice, such as TRPO and PPO. Our results show that, for any parametrization, the convergence rate of the algorithm benefits from an increasing stepsize or, equivalently, from a decreasing weight for the regularization term in the mirror descent update, shedding light on ways to improve the state of the art in policy gradient methods (TRPO/PPO). In fact, the success of TRPO and PPO has largely been attributed to the fact that their updates make sure that the new policy is close to the old one. Our paper suggests that these bounded updates are not necessary and that, on the contrary, larger updates could improve convergence speed.


**[Error floor]** The additional factor $\nu_\mu$ in the error floor is introduced due to the difference in analysis. In fact, the analysis of [Agarwal et al., 2020] relies on the smoothness of $\log\pi_\theta(s,a)$ (or equivalently the boundedness of $\left\lVert \phi(s,a)\right\rVert$), on the boundedness of $\left\lVert w \right\rVert$ and on a stepsize that is decreasing in the number of iterations. On the contrary, our analysis does not depend on any assumptions on the boundedness of $\left\lVert \phi(s,a) \right\rVert$ and $\left\lVert w \right\rVert$ and relies on the performance difference lemma (1) and the three point descent lemma (4). The distribution mismatch coefficient $\nu_\mu$ is also a crude bound on $\nu_k$ introduced at the end of page 8, which is close to 1 when $\pi^t$ is close to $\pi^\star$. Furthermore, additional factors like distribution mismatch/concentrability coefficients or $1/(1-\gamma)$ are present, to the best of our knowledge, in all works on linear convergence analysis, e.g. [Mei et al., 2020; Cayci et al., 2021; Cen et al., 2021; Zhan et al., 2021; Bhandari and Russo, 2021; Xiao, 2022], for both the tabular and log-linear policy settings.

**[Distribution mismatch coefficient]** The distribution mismatch coefficient is a standard element in the policy optimization literature [Agarwal et al., 2021; Xiao, 2022]. In particular, it can be upper bounded by $\frac{1}{(1-\gamma)\min_s \mu(s)}$ and can be equal to $\frac{1}{1-\gamma}$ when we choose $\mu = d_\mu^\star$, which is possible within our framework.

---

### Author Response · Authors · 2022-12-01
**Follow up**

Dear Reviewers,

Just a quick follow up regarding our response to your comments. With our reply, we believe we have addressed all concerns/comments that have been made. As we have not received any additional comment in the meantime, and as the ratings have remained the same, we would like to kindly ask if the Reviewers could to update their reviews in light of our response (and we would be happy to provide additional clarification should there still be any other outstanding concerns).

Best,
The authors

---

### Decision · Program_Chairs · 2023-01-20

**Decision:**

Reject

**Justification For Why Not Higher Score:**

The weakness section above gives somewhat succinct answer.

**Justification For Why Not Lower Score:**

N/A

**Metareview: Summary, Strengths And Weaknesses:**

There has been a lot of recent work on the convergence of policy gradient methods. [Agarwal et al., 2021] provide a very thorough analysis of natural policy gradient (NPG), which divides error into three parts:
1) Optimization error -- which vanishes as more gradient updates are conducted
2) Statistical error -- which vanishes as more samples are used for estimation
3) Bias due to error in the approximation architecture.

Agarwal et. al gave a sublinear convergence rate for the optimization error. Subsequently, several papers point out that this could be made linear when stepsizes are large, at least for tabular representations which avoid issue (3). The idea is that NPG with large stepsizes is policy iteration, which converges linearly. The current submission particularly builds on [Xiao, 2022], who show linear convergence of NPG in the tabular case.

*Strengths*: Linear convergence is not surprising. When there is no error in function approximation and step-sizes are large, NPG essentially   reduces to policy iteration. Still, the literature lacks results that provide policy iteration style convergence rates when large stepsizes are used AND one assumes approximation errors are uniformly small (in an appropriate sense). This paper closes the gap.

Most interesting, perhaps, is that the term capturing limiting bias depends on a certain maximal mean squared error rather than infinity norm error. To my reading, the main trick in the paper is to strengthen an assumption in Agrawal et. al so that large stepsizes are possible while still controlling error. (The authors do not call call this out and instead write that they make a similar assumption...)

*Weaknesses:*  Generalizing NPG with tabular representations to NPG with accurate function approximation is not as substantial as it might first appear: with perfectly accurate approximation, the two algorithms are identical. It feels a bit as if someone stitched Xiao [2022] together with the control of error terms in Agrawal et. al [2022]. All five reviewers worry that the novelty relative to past work is limited. All feel the paper is below the bar. If theoretically minded reviewers who are actively interested in this topic feel this way, it is hard to argue that the work should be highlighted to the broader ICLR community.